# Knowledge Development Trajectories of Crime Prevention Domain: An Academic Study Based on Citation and Main Path Analysis

**DOI:** 10.3390/ijerph191710616

**Published:** 2022-08-25

**Authors:** Song-Chia Hsu, Kai-Ying Chen, Chih-Ping Lin, Wei-Hao Su

**Affiliations:** 1College of Management, National Taipei University of Technology, Taipei 10608, Taiwan; 2Department of Industrial Engineering and Management, National Taipei University of Technology, Taipei 10608, Taiwan

**Keywords:** crime prevention, crime control, environmental conditions and crime, criminal behavior and the law, main path analysis

## Abstract

This study performed main path analysis to explore the academic field of crime prevention. Studies were collected from the Web of Science database, and main path analysis was used to analyze the studies and identify influential authors and journals on the basis of the g-index and h-index. Cluster analysis was then performed to group studies with related themes. Wordle was used to output keywords and word clouds for each cluster, both of which were used as reference to name each cluster. Five clusters were identified, namely crime displacement control, crime prevention through environmental design, developmental crime prevention, the effects of communalism on crime prevention, and the effect of childhood sexual abuse on crime. Each cluster was analyzed, and suggestions based on the results are provided. The main purpose of crime prevention is to advance our understanding of the psychological criminal mechanisms (i.e., personal, social and environmental impacts) associated with different criminal behaviors at the intersection of law by using main path analysis.

## 1. Introduction

Crime prevention has garnered attention worldwide. It refers to the direct and indirect measures taken to reduce, inhibit, and prevent crime. Scholars have analyzed crimes such as theft, juvenile crimes, and robbery and factors related to them to develop crime prevention strategies; however, most crime prevention-related literature reviews have been limited. To comprehensively understand crime prevention, this study performed a main path analysis on the literature on crime prevention, investigating the development history and trends of crime prevention. The objectives were as follows:To identify the overall development trajectory of crime prevention and key studies on crime prevention in the academic field through a main path analysis and find out the relevant research in the academic field of crime prevention in different periods.To identify key themes in the research on crime prevention through a cluster analysis.To identify key themes in the research on crime prevention for different periods through text mining and growth curve, and the growth trend of each research topic is predicted through growth curve analysis.

### 1.1. Crime Prevention Theories

Crime prevention involves improvements in the social and physical environments before, during, and after the occurrences of crimes to prevent, control, eliminate, and reduce them. Criminology approaches crime from the perspective of etiology and the environment [1]. Etiology is used to explore motivations to commit crimes, which are often diverse and cannot be explained by a single theory. As a result, scholars of criminology have combined several aspects of criminology to comprehensively explain these motivations and crimes. Analyzing crimes from the perspective of environment requires framing crimes as trade-offs between costs and benefits and acknowledging that humans’ understanding of consequences informs their decision to commit crimes. Before committing crimes, rational individuals assess the risks, the severity of the penalties, and the rewards. If the rewards outweigh the risks, the individuals will likely commit the crimes. We found that white-collar crime is the cause of crime that has been neglected in academic research. The concept of white-collar crime has a lot of confusion and controversy. The researchers suggest collecting various documents and data packages in recent years, including field observations, interviews and questionnaire surveys and other empirical research. However, in recent years, medical staff has been threatened with verbal threats or even physical violence during the medical process, which not only affects their health or safety but also interferes with medical operations and affects other patients’ treatment. This article used the situational crime prevention theory to prevent medical violence and analyzes the reasons, objects, places, and time of medical violence that often occur in court judgments and medical staff’s perceptions. By increasing the difficulty of crime, increasing the risk of crime, reducing crime opportunities and reducing post-criminal remuneration, we try to develop strategies to prevent medical violence. Through cross-field team cooperation to conduct a case review and preventive strategy management, we will continue to improve the personnel’s ability to prevent and handle violent incidents, and we will strive to maintain the safety of the medical environment and anti-violent incidents. Crime prevention models [2] can be technology-oriented (e.g., punitive, corrective, and mechanical), target-oriented, stage-oriented (e.g., prevention during intention, preparation, and attempt stages), those proposed by Podolefsky (involving strategies analyzing social problems and preventing victimization), conscious strategies (e.g., conservative, liberal, and radical models), those categorized by Naudé (e.g., biopsychic, sociological, physical, and legal sanction and punishment models), and those classified by Clarke (e.g., root cause, deterrence, rehabilitative treatment, and situational models). In conclusion, the model of crime prevention, literature research, and scholars suggest that the concept of “community policing” was born for the police. The local police must actively promote related measures, and the problem determination must show the importance of professionalism to the police. The concept of crime prevention and public service is increasingly advocated, and the police are expected to provide a safe and secure living environment.

#### 1.1.1. Purpose of Crime Prevention

Police enforcement agencies operate under the “tertiary prevention” principle advanced by public health agencies, emphasizing that prevention is better than the cure. Crime prevention involves eliminating opportunities to commit crimes, situational prevention, and relapse prevention. Targeted policies for each of these aspects may be more effective and thus reduce national and social costs. Crime prevention policies are a crucial aspect of national governance and can also be grouped as follows [3]:Primary prevention strategies: to eliminate opportunities to commit crimes by improving physical and social environments.Secondary prevention strategies: to pre-emptively intervene in the activities of potential criminals.Tertiary prevention strategies: to provide treatment to criminals to stop them from committing crimes.

Reviewing the literature on the definition of “tertiary prevention” related measures of prevention, it is particularly recommended that future research scholars put forward more suggestions on preventing child sexual abuse cases, including protecting the identity of the whistleblower and the right to work as well as creating a friendly campus and resettlement agencies environment, so that children have the courage to ask for help.

#### 1.1.2. Types of Crime Prevention

Crime prevention involves individual and situational models. In the criminal justice system, crime prevention involves patrol and arrest in traditional policing, prosecution, quick decisions made by judges, and deterrence and long-term incarceration in rehabilitation institutions. Drug crime prevention involves raising awareness of the risks of drug use, preventing smuggling, investigating drug trafficking and related personnel, and combatting drug trafficking organizations. Situational crime prevention involves measures to decrease the risks of certain specific crimes, reduce the rewards of crimes, and eliminate opportunities to commit crimes through environmental design and management. Community crime prevention involves altering the structure of a community (e.g., family, peers, and neighborhood organizations). Developmental crime prevention involves preventing individuals from developing the intention to commit crimes by eliminating dangers to individual growth, prioritizing children’s health and academic performance, and preventing individuals from becoming criminals because they were abused in their childhood [4,5]. Reviewing the recommendations for future research, scholars in the literature especially elaborate on the rehabilitated people who no longer commit crimes and the unfriendly eyes of the public observing the rehabilitated people. Regarding the characteristics of Japan’s rehabilitation practices, it is explained in perspectives of public sector organizations and manpower, private sector types and functions, crime victims-related measures, and crime prevention and recidivism prevention carry forward. Finally, these experiences of Japan compared with Taiwan’s current practices provide feasible advice for the references of planning and promoting future judicial protection policies by the relevant authorities.

### 1.2. Literature on Main Path Analyses

Several scholars have performed path analyses and key-route main path analyses to review studies on science and technology. Verspagen [6], Fontana et al. [7], and Consoli and Mina [8] performed main path analyses to map technological trajectories. Bekkers and Martinelli [9] and Lucio-Arias and Leydesdorff [10] performed main path analyses to analyze technological changes. Bhupatiraju et al. [11], Calero-Medina and Noyons [12], Colicchia and Strozzi [13], Harris et al. [14], Liu et al. [15], Chuang et al. [16], Yan et al. [17], and Su et al. [18] performed main path analyses on literature reviews in various domains. Lee [19] performed a main path analysis to simplify and organize patent verdicts to explore their effectiveness. Lee [20] performed a main path analysis to identify key patent verdicts and explore patent misuse in the 20th century.

## 2. Materials and Methods

### 2.1. Data Source

This study reviewed literature on crime prevention as the basis for a keyword search. The keyword “crime prevention” was used to search the Web of Science (WOS) database on 31 December 2020; the search yielded 1668 results. Duplicates and studies without authors, titles, and publication dates were removed, leaving 1554 studies.

### 2.2. Main Path Analysis

Main path analyses can be used to process large numbers of citation data and review key knowledge in any academic field. Main paths connect the source (starting) and sink (ending) points in the literature. Main paths are identified by measuring the traversal counts of each path through search path count, search path link count (SPLC), and search path node pair [21]. This study referenced Liu and Lu (2012), who used global main paths and key route main paths. SPLC is superior to search path count and search path node pair [22], because it maps knowledge diffusion more effectively. Figure 1 presents the process of main path analysis based on SPLC weights. First, a path in the network is selected. The number of paths connecting source points and nodes to sink points and the number of possible paths connecting sink points to each node are multiplied to obtain the weight of each link.

### 2.3. Basic Statistics Analysis of Journals and Authors

The WOS database provides the journal in which a study was published, the study’s publication date, the journal’s g-index and h-index, and the author’s name and g-index and h-index. The g-index represents the number of citations an article receives, with top articles receiving at least *g*^2^ citations, whereas the h-index represents the number of citations an author receives. This study used these indices as primary and secondary indices, respectively, to assess journals’ and authors’ contributions to their academic fields; the top 20 journals and authors in crime prevention were examined.

### 2.4. Growth Curve Analysis

This study input the total number of studies on crime prevention published each year in the WOS database into Loglet Lab 4 to create a graph; the Y-axis was the number of papers, and the X-axis was the year. The graph enabled us to identify periods of growth and maturity in the field.

### 2.5. Cluster Analysis

A cluster analysis was performed to group studies with similar themes. Each cluster was named on the basis of the studies’ keywords. An edge-betweenness clustering analysis was then performed. First, the betweenness in the network was calculated. Betweenness is the total number of shortest paths that cross the shortest path between any two nodes. Second, the path with the highest betweenness score was removed. If a new citation network was derived, the modularity of the cluster was calculated. If no new citation network is derived, the first and second steps were repeated until all paths were removed. Finally, the group with the highest modularity was identified as the optimal group; modularity was used to compare the strength of links connecting intercluster and intracluster nodes.

### 2.6. Data Mining

The labels for each cluster were input into Wordle to determine the frequency with which each title appeared in the literature and to output a word cloud; prepositions and definite articles were not included in the computation.

## 3. Results

### 3.1. Data Statistics

Figure 2 presents an Excel chart of the number of studies published each year and total number of studies on crime prevention from 1989 to 2020 in dark and light blue, respectively. The number of studies consistently increased. By 2015, the growth was exponential, and hundreds of studies were being published annually. This signifies that crime prevention began to garner more attention.

#### 3.1.1. Crime Prevention Journals

This study used the g-index to identify the top 20 journals covering crime prevention. Journals with the same g-index scores were ranked on the basis of their h-index scores (Table 1). The top five journals, in descending order, are *The British Journal of Criminology*, *Journal of Experimental Criminology*, *Security Journal*, *Journal of Research in Crime and Delinquency*, and *Criminology & Public Policy*. The top journal, *The British Journal of Criminology*, published its first issue in 1989 and has since published 75 studies. The journal publishes high-quality studies on crime prevention from around the world and has indicated that professionals in criminology, sociology, anthropology, and psychology are crucial to crime prevention. *Journal of Experimental Criminology* focuses on criminological theories and high-quality experimental research for the development of judicial policies. *Security Journal* mostly consists of security analyses and reports on research and innovations in various fields. *Journal of Research in Crime and Delinquency* provides research notes, reviews of other journals, and explorations of controversial problems in the judicial field. *Criminology & Public Policy* centers on judicial policies and their implementation. CiteScore is based on the citations recorded in the Scopus database rather than in JCR, and those citations are collected for articles published in the preceding four years instead of two or five. Using the CiteScore of Scopus to rank journals, we found that *Annals of the American Academy of Political and Social Science* and *Criminology* were ranked first and second, respectively. Either *g*-index or CiteScore provide researchers with a measure of the journal, and it will be referenced.

#### 3.1.2. Authors

This study used the g-index to identify the top 20 authors in crime prevention. Authors with the same g-index score were ranked on the basis of their h-index scores (Table 2). The top six authors were Welsh, BC; Farrington, DP; Weisburd, D; Braga, AA; Johnson, SD; and Piza, EL.

BC Welsh is a professor in the Northeastern University Criminology and Criminal Justice Program and serves as the director of the Cambridge Somerville Youth Study. Welsh’s main field of research is crime prevention, and he has served as an advisor for the United Nations Office on Drugs and Crime, the United Kingdom Home Office, and the Canadian National Police Service. Six of the 18 studies in the global main path, accounting for one-third, were written by Welsh, indicating his contributions to the field.

DP Farrington is a British criminologist, forensic psychologist, and emeritus professor of psychological criminology at the University of Cambridge, where he also serves as a Leverhulme Trust Emeritus Fellow. He is renowned for his research on the development of criminal behavior. In 2003, Farrington was appointed an Officer of the Most Excellent Order of the British Empire for his contributions to the field of criminology. The same year, he received the Stockholm Prize in Criminology for his contributions to crime prevention programs.

D Weisburd is an Israeli criminologist and Distinguished Professor of Criminology, Law, and Society at George Mason University. Weisburd’s main fields of research are public security and white-collar crime. In 2010, he received the Stockholm Prize in Criminology and Israel’s Rothschild Prize in social sciences.

AA Braga is a distinguished professor of criminology and criminal justice. He is renowned as a leading researcher in crime prevention and has published various influential papers in *Criminology*, *Journal of Research in Crime and Delinquency*, *Justice Quarterly*, and *Journal of Quantitative Criminology*.

SD Johnson is the Director of the Dawes Center for Future Crime at University College London. He works in criminology and forensic psychology, and his research focuses on how new technologies affect crime.

Elis Piza is an Associate Professor at the John Jay College of Criminal Justice of the City University of New York. He has conducted research on geographic information systems and spatial analysis. He also published six of the studies in the global main path, which is a similar contribution to that of Welsh.

### 3.2. Academic Literature and the Overall Development Trajectory of Crime Prevention

The main path analysis revealed the global main path of crime prevention in the academic field. In Figure 3, the green and blue nodes represent the source and sink points, respectively. Each node represents a study. The arrows connecting the nodes indicate the direction of knowledge flow. Each node was labeled with a serial code corresponding to the last name of the first author, the other authors’ initials, and the year of publication. If two studies had the same serial code, a lowercase letter was added to the end of the code to indicate which study was first alphabetically.

Figure 3 depicts the global main path of crime prevention in the academic field. The main path of the citation network had the highest total weight and consisted of 18 nodes, each representing a study. Pennell et al. [23] assessed the effects of the Guardian Angels Program (i.e., volunteers patrolling the streets and subways) on crime and citizens’ attitude toward crime; order maintenance policing helped prevent crime and made citizens feel safe. Bennett and Lavrakas [24] proposed a community-based crime prevention program to eliminate the fear of crime and other problems but did not observe any changes in crimes or perceived quality of life. Mulvey et al. [25] reported that interventions must be broadly based and assessed more comprehensively to deter juvenile delinquency. Graham and Bennett [26] explored situational and community crime prevention programs in Europe and North America and their implementation effects.

Some scholars have assessed the economic efficiency of crime prevention programs. Welsh and Farrington [27] evaluated the feasibility and economic efficiency of a developmental and situational crime prevention program and explored its ability to reduce crime. Welsh and Farrington [28] proposed a cost–benefit analysis method to determine the optimal approach to crime prevention. The focus for most scholars of crime prevention has shifted to the effects of crime reports on crime rates. Painter and Farrington [29] indicated that increasing street lighting reduced crime rates on the basis of a self-report survey of young people. Farrington and Welsh [30] recommended that studies measured crime rates by using police records, surveys of victims, and accounts of offenses by perpetrators.

The widespread use and popularity of closed-circuit television (CCTV) have prompted scholars to explore the effects of surveillance cameras on crime. Welsh and Farrington [31] reported that CCTV reduces crimes in parking lots considerably but does not affect violent crime. Welsh and Farrington [32] evaluated the effects of CCTV as an intervention for crimes and revealed that it was more effective in the United Kingdom than in the United States; however, the effect was limited to crimes in parking lots. On the basis of survey results, Piza et al. [33] reported that CCTV should be placed in areas with high crime rates to discourage crimes. When installing CCTV, factors such as criminals and the presence of ground obstacles should be considered.

Piza and O’Hara [34] introduced a project that combined CCTV and foot patrols. Although the number of murders and shootings decreased in areas with foot patrol, the number of robberies exhibited temporal and spatial displacement. Piza et al. [35] indicated that multiple interventions should be implemented simultaneously to prevent crime, especially street crime. Piza et al. [36] presented an intervention combining CCTV monitoring with directed patrolling that successfully reduced crime rates.

Piza [37] used propensity score matching to explore the effects of CCTV on auto thefts, auto parts thefts, and violent crimes and found that CCTV only deterred auto thefts. Welsh et al. [38] noted that private CCTV systems are more effective in preventing crimes than are police CCTV systems or the combination of police CCTV systems and security personnel. Khan et al. [39] proposed adding blockchain to surveillance systems to ensure video evidence cannot be compromised. Most early studies focused on planning and assessing crime prevention strategies involving physical infrastructure; only in 2003 did studies appear related to CCTV.

## 4. Discussion

### 4.1. Development Trajectory of Crime Prevention and Clusters

The key-route main path comprised 23 studies. A comparison between Figure 3 and Figure 4 indicated that the key-route main path contained all 18 studies in the global main path; thus, the studies in the global main path are crucial in the academic field of crime prevention. The key-route main path also contained five studies not in the global main path. Polvi et al. [40] explored the risks of being burglarized a second time and revealed that homes burglarized once in 1987 had a four times greater risk of being burglarized again in the same year. Trickett et al. [41] analyzed the differences between areas with high and low crime rates and revealed that vulnerability determined the risks of crimes; thus, crime prevention strategies should focus on the most vulnerable populations and places. Farrell and Pease [42] performed predictable seasonal variation analyses by using police calls to domestic disputes and residential burglary in a 3-year period, and they found that larger predictable seasonal variations may provide deeper insight into the problems in question and directions for crime prevention. Farrington [43] categorized criminology research into five categories, namely measurement, explanations or correlates, police or crime prevention, sentencing, and correctional treatment, and they suggested that long-term research on the causes of crime is warranted. Guerette and Bowers [44] evaluated situational crime prevention projects and reported that crime displacement entailed the diffusion of the benefits of an intervention; this indicates that certain interventions prevent crimes.

### 4.2. Cluster Analysis of Field of Crime Prevention

An edge-betweenness cluster analysis yielded 20 clusters, and studies in the top 5 clusters, namely the effects of crime displacement control on crime prevention, the effects of crime prevention through environmental design (CPTED), the effects of developmental programs on crime prevention, the effects of communalism on crime prevention, and the effects of childhood sexual abuse on crimes, were input into Wordle to identify keywords. Figure 5 presents the themes, number of studies, keywords, and word clouds for these clusters. Keywords were ranked by their frequency (numbers in parentheses) in titles; for example, “crime” appeared an average 0.33 times in the first cluster. The studies in each cluster were analyzed to determine the main path of each cluster and the research direction of each main path (Figure 6, Figure 7, Figure 8, Figure 9 and Figure 10). The literature growth trend charts revealed that the number of studies in the literature increased in all five clusters.

Figure 5 translation: (top row, left to right): research theme; first cluster (122 studies): the effects of crime displacement control on crime prevention; second cluster (104 studies): the effects of crime prevention through environmental design; third cluster (83 studies): the effects of developmental programs on crime prevention; fourth cluster (59 studies): the effects of communalism on crime prevention; fifth cluster (53 studies): the effects of childhood sexual abuse on crimes. (Bottom left, top to bottom): number of studies; word clouds.

#### 4.2.1. The Effects of Crime Displacement Control on Crime Prevention

The first cluster contained 122 studies on crime displacement control (Figure 6), and the main path contained 11 studies on crime displacement control from 1990 to 2020, demonstrating the effects of crime displacement control on crime prevention.

Gabor [45] indicated that citizens value community-based crime prevention programs and public participation. Braga [46] suggested that focused police actions can prevent crimes and disorder in crime hot spots. Weisburd et al. [47] indicated that focused crime prevention efforts at target sites improve nearby areas. Braga and Bond [48] reported that situational prevention strategies are more effective than misdemeanor arrests in preventing crime. Durlauf and Nagin [49] stated that the certainty of punishment deters criminals.

Johnson et al. [50] proposed the EMMIE framework, which involves collecting primary data on crime prevention and synthesizing knowledge through systematic reviews. Santos and Santos [51] explored police- and family-oriented interventions and discovered that their effects on crime were nonsignificant. Braga et al. [52] found that the effects of focused police crime prevention interventions in crime hot spots result in a diffusion of crime control benefits; the results showed that hot spots policing diffused crime control benefits into proximate areas. Piza et al. [53] reported that police substations reduce the number of thefts and scooter thefts in target areas considerably.

#### 4.2.2. CPTED

The second cluster consisted of 104 studies on CPTED, which involves the relationship between environmental characteristics and crimes (Figure 7). Taylor [54] maintained that involvement in the community encourages residents to coordinate their efforts against unrest and ensure social stability. Loukaitou-Sideris [55] explored bus stop crimes and analyzed the environmental attributes that determine safety for bus drivers. The study also developed a crime prevention strategy for bus stops based on environmental attributes. Loukaitou-Sideris and Eck [56] investigated whether crimes are a barrier to active living and developed a theoretical model comprising situational characteristics, crime and disorder, fear of crime or disorder, and physical activity.

Marzbali et al. [57] proposed CPTED and explored its effects on burglary rates. Marzbali et al. [58] explored the effects of CPTED on victims and fear of crime. Abdullah et al. [59] reported that elderly individuals perceive higher levels of neighborhood cohesion than do younger individuals. Marzbali et al. [60] examined the effects of CPTED on residential burglary rates, and they concluded that CPTED was associated with low victimization rates. Chen et al. [61] analyzed the effects of floating populations on residential burglary; floating populations from a single province do not strongly affect residential burglary rates in most parts of the city, whereas those from multiple provinces significantly and positively affect residential burglary rates. Cho and Jung [62] reported that CPTED is an effective tool for preventing sexual harassment and that although crime displacement occurs, its effects are outweighed by the positive effects of new policies in target zones. He et al. [63] analyzed factors affecting the safety of traditional settlements and revealed that safety is spiritual, physical, and behavioral. Kim and Park [64] investigated three groups’ awareness of wall removal projects. The first group consisted of burglars stationed at houses without walls, the second group consisted of residents who were stationed at houses with walls, and the third group consisted of residents who were stationed at houses without walls. The first group perceived houses without walls as easier targets, whereas the second and third groups reported that wall removal increased their fear of crimes. Most of the residents indicated that the walls protected them from external threats and ensured their environment was safe.

#### 4.2.3. The Effects of Developmental Programs on Crime Prevent

The third cluster consisted of 83 studies on developmental crime prevention (Figure 8). Mackenzie [65] used rigorous science scores to assess crime prevention strategies. Welsh and Farrington [66] commented that the effectiveness of correctional strategies should be evaluated and that the results of the evaluations should be incorporated into policies and practices. Manning et al. [67] encouraged policymakers to make rational decisions and incorporate research evidence into policies.

Crowley [68] encouraged policymakers to consider their regional planning and implementation capacity when investing in crime prevention programs. Farrington [69] applied a universal index to juvenile crime to identify suitable strategies for preventing each type of crime and revealed that family interventions and preschool courses can reduce juvenile crimes. Weisburd et al. [70] performed a systematic review of crime prevention strategies and suggested that studies perform cost–benefit analyses and explore the effectiveness of each strategy. Spasic and Simonovic [71] compared police officers’ and police educators’ receptivity to evidence-based policing. Farrington et al. [72] invited six leading criminologists to discuss instability in social systems, the importance of systematic reviews, heterogeneity among studies’ conclusions, replicating experiments in procedural justice, and enthusiasm bias in criminal justice experiments. Picasso and Cohen [73] employed a survey methodology previously used in studies on the environment, health, and safety economics to estimate the costs of violent crimes and homicide and demonstrated the feasibility of the methodology.

#### 4.2.4. The Effects of Communalism on Crime Prevention

The fourth cluster consisted of 59 studies on crime prevention and communalism (Figure 9). King [74] detailed the relationship between France’s crime prevention strategies and antisocial behavior. Omalley [75] reported that the rise of neoconservatism discouraged the development of programs based on risk models and examined situational crime prevention in relation to neoconservatism. Omalley [76] defined crime prevention as a key responsibility, advocated for the causes of crime to be studied, and indicated that criminals should take responsibility for their crimes. The study also proposed strengthening the law to enhance its punitive function.

Omalley et al. [77] believed that overly political governmentality may impede policy implementation and advocated for public discussion of this notion. Carson [78] explored the effects of communalism on crime prevention strategies. Hughes [79] examined community crime prevention, the instability of community governance, and strangers’ effects on community safety. Carson [80] expanded on Carson [80] by continuing to explore communalism as well as feasible crime prevention strategies. Selmini [81] explored crime prevention and social reassurance in Italy, which are strongly affected by conflict and negotiations between Italy’s national and local governments. Hughes [82] combined international, national, and local strategies to address changing governance in a society attempting to regulate migration and ensure community safety.

#### 4.2.5. The Effects of Childhood Sexual Abuse on Crime Prevention

The fifth cluster consisted of 53 studies on crime prevention and childhood sexual abuse (Figure 10). Beauregard et al. [83] identified three hunting process scripts in serial sex crimes, namely coercive scripts, manipulative scripts, and nonpersuasive scripts. LeclercPB2009 [84] analyzed the victims of various crimes to identify specific characteristics in potential victims. Leclerc et al. [85] proposed a script model for child sex offenses.

Leclerc et al. [86] examined the effects of guardians on the severity of childhood sexual abuse; the presence of a guardian decreases the duration of sexual abuse by 86%, suggesting that guardians are crucial in preventing sexual crimes. Leclerc et al. [87] reported that environmental criminology contributes to the development of strategies to prevent childhood sexual violence. The study collected information related to sexual crimes through questions posed by criminologists and environmental criminologists and offered directions for further research. Guerzoni [88] evaluated the feasibility of clergy’s child safety practices to minimize impropriety and protect children from sexual crimes.

From 2007 to 2011, most studies focused on childhood sexual abuse; after 2015, the research focus shifted to strategies for preventing such crimes (Figure 10).

### 4.3. Crime Prevention Growth Curve Analysis

The sample comprised 2678 scholars in the academic field of crime prevention. A growth curve analysis was performed to identify the period of maturity in the research on crime prevention. The dotted line in Figure 11 represents the estimated number of papers in the field; the line overlaid with dots represents the actual number of papers. The inflection point of the growth curve was estimated to be 2017, and the field is expected to reach maturity by 2047. By then, the total number of papers is expected to be 2546. At the date of writing, the crime prevention field has passed the inflection point and is expected to slowly enter the maturity period.

## 5. Conclusions

This study collected 1554 studies on crime prevention, which is a field expected to reach maturity by 2047, at which point 2546 studies are expected to have been published. This study performed path analyses to identify the path with the highest total weight and examined it with respect to the key-route main path. Early studies focused on strategies to prevent various types of crime, middle-stage studies centered on cost–benefit analyses of crime prevention strategies, and late-stage studies explored the effects of physical infrastructure on crime rates. The following are three periods.

Early-term (1989–1997): Proposed prevention programs for different crime types. In order to improve social order and reduce citizens’ fear of crime, build a mutual aid and security system to reduce community-based crime, use crime propaganda to promote campus crime to protect young people, and reduce the causes of crime.Mid-term (1998–2002): Substantial effectiveness evaluation of situational prevention programs. Through crime reporting and cost–benefit analysis, adding street lights can indeed reduce crime rates. This phase is also aimed at reducing the incidence of youth crime, with youth self-report studies being a valid and reliable measure.Later period (2003–2020): Discuss the impact of physical construction and crime reporting on crime rates. Through the active monitoring system and the scheme of cooperating with the police patrol, the security guards and the police mixed the operation of the monitor and the use of blockchain technology to ensure that the stored video is not modified and the data are authentic.

This study also performed a cluster analysis and data mining and identified the following five clusters, which were used to further analyze the evolution of the literature:The effects of crime displacement control on crime prevention: There are 122 articles in this group. In the early period (1996–2014), there were 87 literature reviews in the study, and studies examined focused police action. The most discuss police roles, police executive powers and applications, and integrity organization design. In the late stage (after 2015), there were 35 literature reviews in the study, and studies evaluated the effects of crime displacement control and other strategies. A more systematic solution is to collect relevant information and raw data (such as transportation, community monitoring, formal cases); the acceleration of big data application is certainly the crucial strategy of scientific development in Taiwan at this stage. It is hoped that the application of big data would enhance the governmental efficacy and articulate examining public need.The effects of CPTED: There are 104 articles in this group. Early (1996–2007) studies focused on the relationship environmental characteristics and social disorder, and late-stage (after 2011) studies examined CPTED. A total of 28 articles were reviewed discussing the impact of burglary and sexual assault on crime prevention. Among them, 15 articles discussed the phenomenon of homeowners’ fear of burglary crimes. Neighborhood cohesion can reduce the rate of residential burglary; seven articles reviewing the literature discussing reducing sexual assault in target areas. It is particularly pointed out that potential high-risk offenders with previous convictions are under the government’s management, and they are also the main benefactors of current social safety net counseling and the construction of social prevention safety nets; six studies in the literature discuss the protection of outer walls, the fear of crime and the function of keeping yourself in a safe environment.The effects of developmental programs on crime prevention: There are 104 articles in this group. Early (2000–2013) studies explored the relationship between policy and crime prevention; the main research direction is the use of scientific methods and inclusion in policy making and enabling decision makers to put research evidence into the policy making process more rationally. In the late stage (after 2014), studies analyzed the effects of developmental crime prevention programs. A total of 36 articles in the literature discuss the feasibility of constructing a common indicator and cost–benefit analysis evaluation and estimating crime prevention costs.The effects of communalism on crime prevention: There are 59 articles in this group. Early (1989–1994) studies mainly researched the nature of crime prevention, whereas late-stage (after 2014) studies explored the effects of communalism on crime prevention. Reviewing 25 studies in the literature, the topics discussed are all about the substantive effect of proper government governance and community security on crime prevention.The effects of childhood sexual abuse on crimes: There are 53 articles in this group. Early (2007–2011) studies analyzed childhood sexual abuse; the literature discusses the connotation and related research of boundary trauma, self-trauma and complex post-traumatic stress reaction caused by abuse. In contrast, late-stage (after 2015) studies explored programs for preventing such abuse.

The global main path analysis, key-route main path analysis, and cluster analysis revealed that the field of crime prevention contains various interrelated themes. The following six major topics all appeared on the global main path and key-route main path, but only three topics appeared in the five groups.

The themes from the global main path and key-route main path, namely cost–benefit analyses of crime prevention strategies, corresponded to the themes identified through cluster analysis, namely the effects of CPTED. We finds that there is a growing body of scientific research that shows that early prevention is an effective and worthwhile investment of public resources.“Exploring the Impact of Physical Construction and Crime Reporting on Crime Rates”, “Exploring the Substantial Effects of Surveillance on Crime Prevention”, “Exploring the Impact of the Integration of Surveillance and Police Patrols on Crime Rates” and “Exploring the Effects of Surveillance on Crime Rates” impact are four issues that can be classified into the cluster analysis “Assessing the impact of development programs on crime prevention”. Looking back at the literature review of the past two years, the concept of smart buildings has been applied to security measures for crime prevention, adding smart street lights and joint patrols by the police and the public to eliminate target areas such as communities or campuses that are prone to crime.The “crime prevention methods proposed for different crime types” can correspond to the influence of crime diversion control on crime in the cluster analysis. Looking back at the literature review of the past two years, the majority of crime types are juvenile deviant behavior and online misconduct. The results show that deviant behaviors of adolescents are affected by law cognition and friend support, and online misbehaviors of adolescents are affected by law cognition and welfare needs. Accordingly, schools should provide law-related courses and information to young people to avoid deviant behaviors and online misbehaviors, so that the youth understand the consequences of their risky behaviors. In addition, young people’s interaction with their peers has to be paid attention in order to avoid peer instigation and participation in deviation behaviors. For youth’s online misbehaviors, it properly links to the youth’s various welfare resources. Practitioners should understand the problems that they encounter in their lives and online spaces, and provide appropriate assistance to reduce their online misbehaviors.

The research framework established by this research and the analysis method chosen are an integrated research method, which is very suitable for application in exploring the technological development trend of a certain topic and expounding the key implications of field management.

## Figures and Tables

**Figure 1 ijerph-19-10616-f001:**
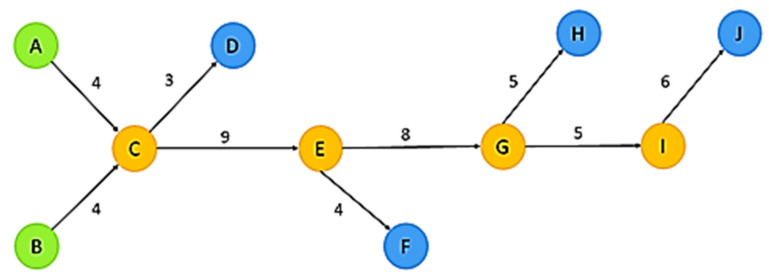
Example of main path analysis based on SPLC weights.

**Figure 2 ijerph-19-10616-f002:**
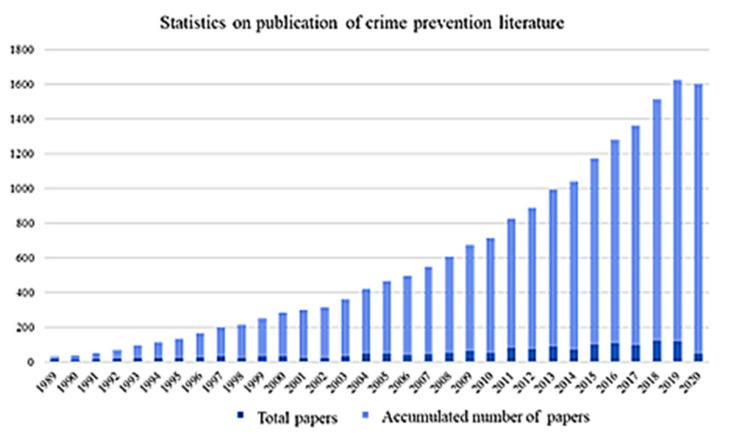
Cumulative papers on crime prevention. Title: Statistics on publication of crime prevention literature. Bottom left text: number of studies published annually. Bottom right text: total number of studies published.

**Figure 3 ijerph-19-10616-f003:**
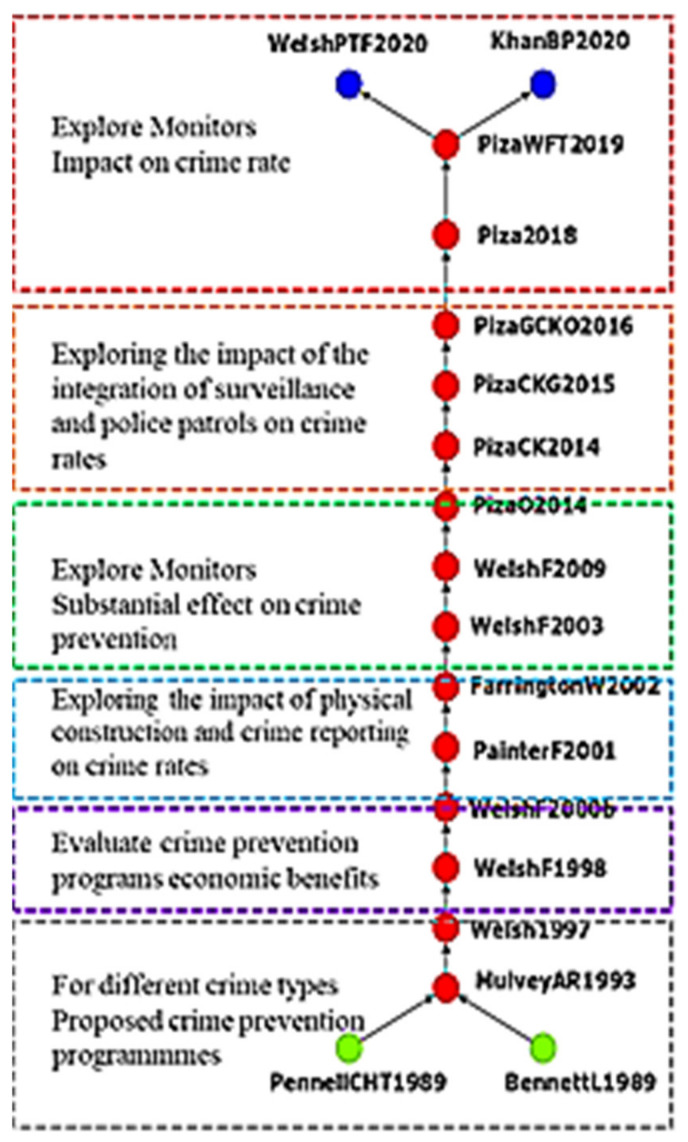
Main path analysis of crime prevention (Top to bottom). Exploring the effects of CCTV on crime rates. Exploring the effects of the combination of foot patrol and CCTV. Exploring the effects of CCTV on crime prevention. Exploring the effects of physical infrastructure and crime reports on crime rates. Evaluating the costs and benefits of crime prevention strategies. Proposing prevention strategies for various types of crime.

**Figure 4 ijerph-19-10616-f004:**
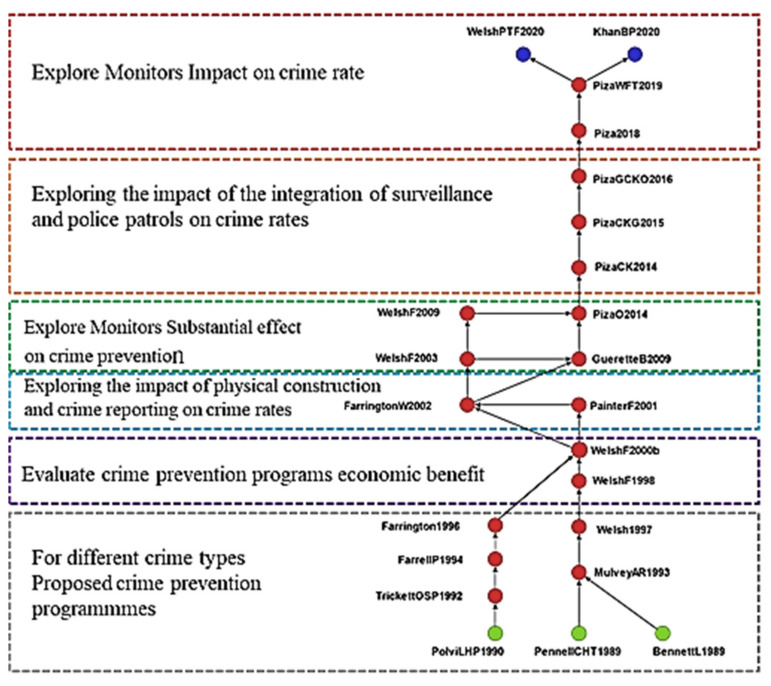
Key-route main path analysis of crime prevention (top to bottom). Exploring the effects of CCTV on crimes. Exploring the effects of interventions combining foot patrol and CCTV on crime rates. Exploring the effects of CCTV on crime prevention. Exploring the effects of physical infrastructure and crime reports on crime rates. Evaluating the costs and benefits of crime prevention strategies. Proposing prevention strategies for various types of crime.

**Figure 5 ijerph-19-10616-f005:**
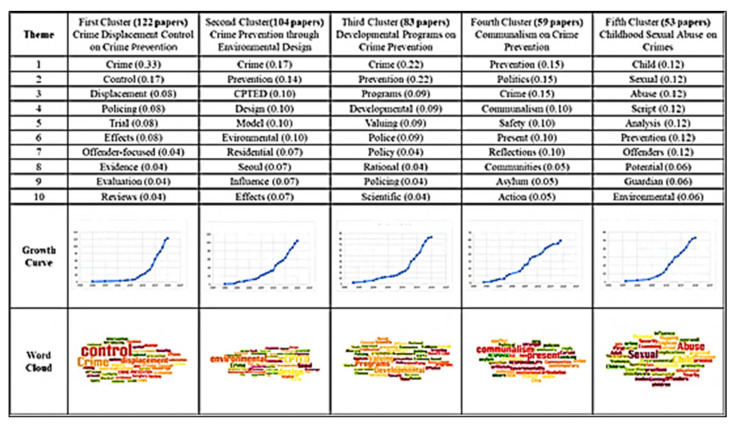
Top 5 clusters of crime prevention-related academic literature.

**Figure 6 ijerph-19-10616-f006:**
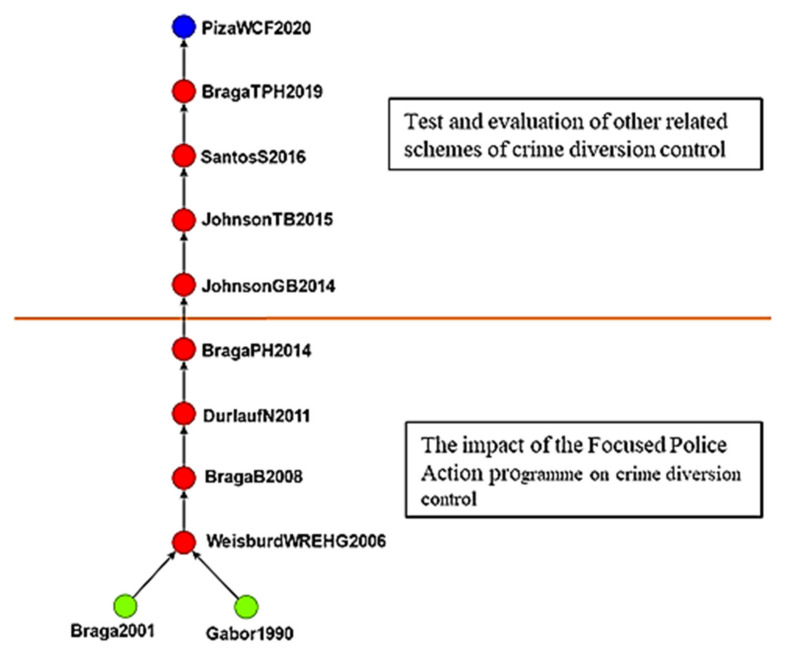
Global main path analysis of first cluster. (Top to bottom). Assessing crime displacement control and other programs; the effects of focused police action programs on crime displacement control.

**Figure 7 ijerph-19-10616-f007:**
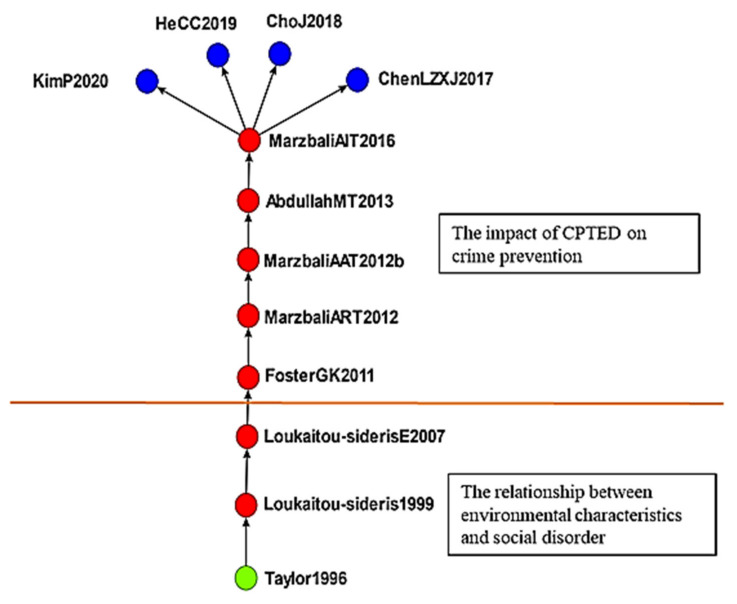
Global main path analysis of second cluster. (Top to bottom). The effects of CPTED; relationship between environmental characteristics and social disorder.

**Figure 8 ijerph-19-10616-f008:**
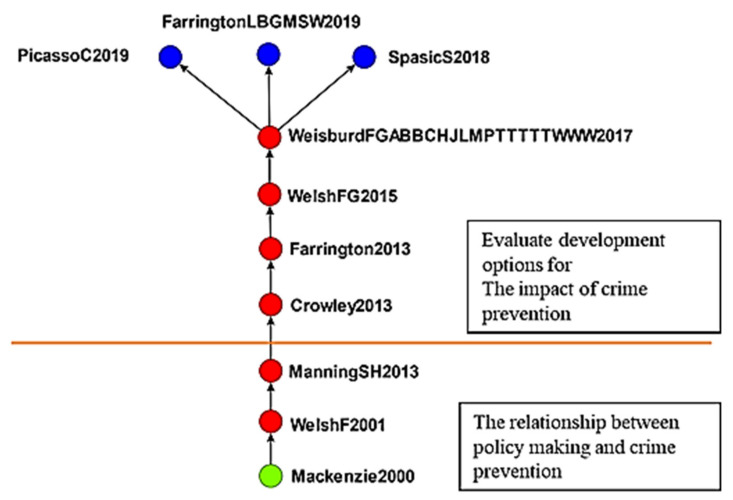
Global main path of third cluster. (Top to bottom). Evaluating the effects of developmental crime prevention; relationship between policy and crime prevention.

**Figure 9 ijerph-19-10616-f009:**
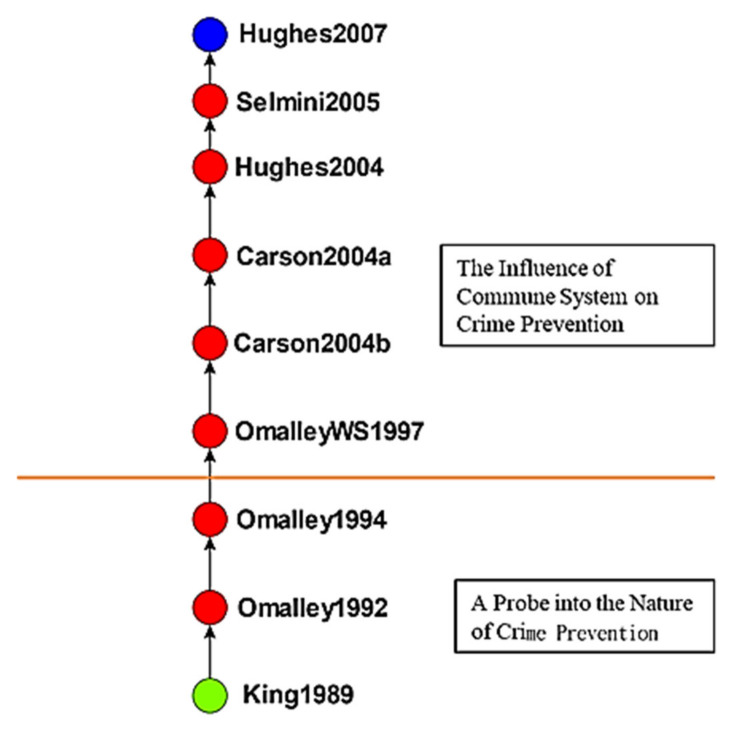
Global main path analysis of fourth cluster. (Top to bottom). The effects of communalism on crime prevention; exploring the essence of crime prevention.

**Figure 10 ijerph-19-10616-f010:**
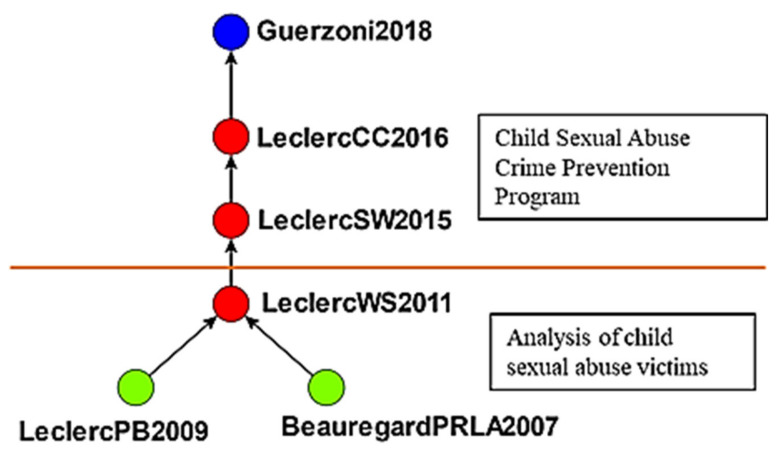
Global main path analysis of fifth cluster (Top to bottom). Strategies for preventing childhood sexual abuse; analysis of childhood sexual abuse.

**Figure 11 ijerph-19-10616-f011:**
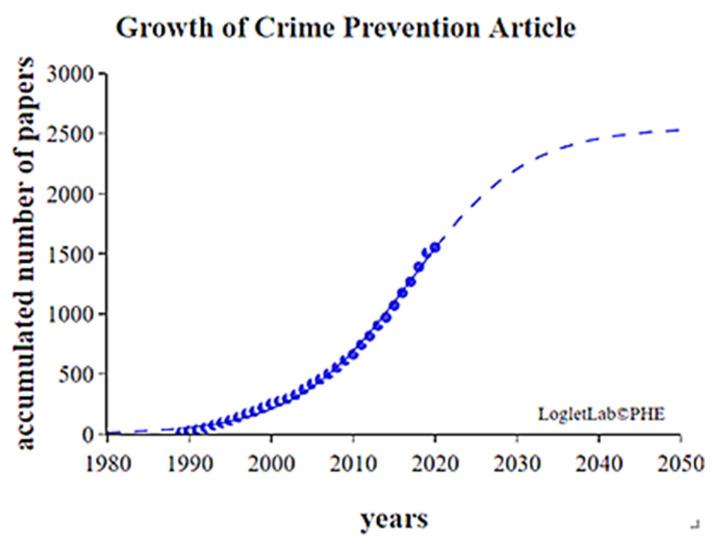
Studies on crime prevention. A growth curve analysis was performed to identify the period of maturity in the research on crime prevention as blue dots.

**Table 1 ijerph-19-10616-t001:** The top 20 most influential journals.

g-Index	Journal	g-Index	h-Index	CiteScore	Active Years	Total Papers	Papers after 2000
Ranking							
1	*British Journal of Criminology*	40	23	5.0	1989~2020	48	75
2	*Journal of Experimental Criminology*	24	12	4	2009~2019	30	30
3	*Security Journal*	23	13	2.3	2008~2020	79	79
4	*Journal of Research in Crime and Delinquency*	23	12	4.8	1993~2020	22	23
5	*Criminology & Public Policy*	23	11	5.6	2010~2020	29	29
6	*Journal of Quantitative Criminology*	21	13	6.8	1996~2019	17	21
7	*Annals of The American Academy of Political and Social Science*	21	9	8.1	1989~2018	14	21
8	*Crime & Delinquency*	20	10	4.4	1989~2020	20	25
9	*Justice Quarterly*	19	11	6.3	2002~2019	19	19
10	*Journal of Criminal Justice*	19	10	5.2	1992~2019	27	31
11	*European Journal on Criminal Policy and Research*	18	10	3.0	2008~2019	32	32
12	*Australian And New Zealand Journal of Criminology*	18	9	2.7	1989~2020	28	36
13	*Crime Law and Social Change*	17	8	5.2	1994~2020	16	22
14	*Canadian Journal of Criminology and Criminal Justice*	16	9	1.5	2005~2019	28	28
15	*Policing-An International Journal of Police Strategies & Management*	15	10	2.5	1999~2019	24	25
16	*Criminology*	14	12	7.3	1997~2019	13	14
17	*European Journal of Criminology*	14	9	3.7	2009~2020	17	17
18	*Theoretical Criminology*	14	9	4.2	2000~2020	14	14
19	*Criminal Justice and Behavior*	13	8	3.2	1991~2019	12	13
20	*Journal of Architectural and Planning Research*	11	6	0.3	1993~2009	9	12
					Total	498	566

**Table 2 ijerph-19-10616-t002:** The top 20 most influential authors.

g-Index	Author	g-Index	h-Index	1st Authors	Active Years	Total Papers
Ranking						
1	Welsh, BC	31	19	26	1997–2020	40
2	Farrington, DP	28	15	7	1996–2020	29
3	Weisburd, D	16	12	11	1997–2019	16
4	Braga, AA	16	10	12	2001–2019	16
5	Johnson, SD	12	8	5	1998–2020	12
6	Piza, EL	11	8	10	2014–2020	15
7	Leclerc, B	11	7	6	2007–2019	11
8	Pease, K	11	6	4	1989–2011	11
9	Eck, JE	10	6	2	2006–2020	10
10	Groff, ER	9	8	3	2007–2019	9
11	Losel, F	9	8	3	2006–2019	9
12	Reynald, DM	9	7	4	2009–2019	9
13	Farrell, G	9	6	4	1994–2018	9
14	Casteel, C	9	5	4	2000–2016	9
15	Andresen, MA	8	6	3	2014–2020	8
16	Beauregard, E	8	6	3	2007–2018	8
17	Bowers, K	8	6	1	2010–2019	8
18	Ekblom, P	8	6	7	1995–2016	8
19	Sherman, LW	8	6	6	1993–2019	8
20	Guerette, RT	7	4	3	2009–2019	7
					Total	252

## Data Availability

Data sharing is not applicable to this article.

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
