# Peer review of "Knowledge Development Trajectories of Crime Prevention Domain: An Academic Study Based on Citation and Main Path Analysis"

_ijerph, 2022, doi:10.3390/ijerph191710616_

Round 1

Reviewer 1 Report

I apologize for the late review, but I enjoyed reading this manuscript very much. This paper's contents are original, and I particularly liked the way the authors analyzed paths and clusters. 

In the manuscript, the quality of the figures is a major concern.  Please provide high-resolution figures in the revised version so I can at least read them and review this paper further. 

Furthermore, journal rankings should not be determined solely by g-scores.  It is well known in the fields of criminology and criminal justice that Criminology is the 1st premier journal, followed by other top journals such as Justice Quarterly and Journal of Quantitative Criminology.  It is vital that authors provide further justification with other journal rankings if they wish to keep the BJC the top foremost journal in this field. 

Reviewer 2 Report

This study performed main path analysis to explore the academic field of crime prevention. It has some interesting information, but there are still some problems in this paper that need further consideration, so as to optimize this research.

1.     The summary of the literatures should not be limited to description, but should point out the shortcomings and future research directions.

2.     This manuscript has no obvious conclusions and does not bring new knowledge. The authors should further refine and summarize the results.

3.     The introduction section needs to be reorganized to clarify the research objectives and research questions.

4.     All the pictures are too blurred, and sharper pictures need to be provided.
